# A Novel UWB Positioning Method Based on a Maximum-Correntropy Unscented Kalman Filter

**Mujie Zhao** [1],*, **Tao Zhang** [1],* **and Di Wang** [2]

1   Key Laboratory of Micro-Inertial Instrument and Advanced Navigation Technology, Ministry of Education, School of Instrument Science and Engineering, Southeast University, Nanjing 211100, China
2   College of Energy and Electrical Engineering, Hohai University, Nanjing 211100, China
*   Correspondence: zhaomujie168@163.com (M.Z.); zhangtao22@seu.edu.cn (T.Z.)

**Abstract:** Aiming at the problem of measurement-information abnormal-error and nonlinear filtering in UWB navigation and positioning, an ultra wideband position algorithm based on a maximum cross-correlation entropy unscented Kalman filter is proposed. The algorithm first obtains the predictive state estimate and the covariance matrix through traceless transformation. Then, it reconstructs observation information using the nonlinear regression method based on the maximum cross-correlation entropy criterion, which enhances the robustness of the unscented Kalman filter algorithm for heavy-tailed noise. The simulation and actual test results show that this algorithm has better positioning accuracy and stability than the traditional filter algorithm in a non Gaussian noise environment. This algorithm effectively solves the problem that UWB indoor location is easily affected by indoor environments, resulting in fixed deviation for that location.

**Keywords:** UWB; UKF; position; maximum correntropy

## 1. Introduction

In recent years, with the advent of the Internet of Things era, all aspects of human life have basically been related to location-based services. For example, in large supermarkets, people can quickly and conveniently find particular goods. In large office buildings, people can find particular locations accurately and quickly. Parents can discern their children's location in real time to avoid losing them. Vehicles in a parking lot can automatically find parking spaces and park. These services are inseparable from accurate location information. Positioning and navigation technology has become the key to smart city construction, which directly affects people's safety and economic development. Therefore, research and implementation of location technology based on wireless technology has an extremely high engineering-application value [1].

The global navigation satellite system (GNSS) is a positioning system based on satellites; it uses satellites as basic reference objects. Because of its high positioning accuracy and its high strategic value, GNSS has now become a focus of all countries and has independently developed its own navigation systems. Compared with other positioning systems, it has unique advantages. Satellite signals can be sent to the global scope, enabling people to obtain positioning information anytime and anywhere. In addition, the satellite system has other characteristics, such as high positioning accuracy, high robustness and all-weather free-space optical communication technology. However, because transmission of satellite signals over a long distance is very weak by the time the signals reach the ground, those signals could rapidly decline or even be unable to be used under the influence of buildings on the ground [2]. Generally, satellite navigation system is suitable for open outdoor environments, while in indoor environments, people urgently need a high-precision, highly reliable indoor positioning system to meet their growing needs.

Ultra wideband (UWB) technology is a new type of wireless communication technology. At this stage, it is mainly used to obtain point-to-point distance based on time and

achieve final positioning with the distances between the label to be tested and multiple reference base stations. Because UWB technology uses narrow nanosecond nonsinusoidal pulses for data transmission and its spectrum range is very wide, in theory, UWB technology can achieve centimeter-level positioning accuracy [3]. In addition, UWB technology also has the advantages of low transmission power, strong multipath anti-interference capability and high security, so it is very suitable for indoor and other closed-area positioning applications.

Non line of sight (NLOS) environments affect the positioning accuracy of UWB technology, and many scholars have put forward their own opinions on reducing NLOS error. A study in the Reference [4], proposed a tight combination of UWB/inertial navigation system (INS) indoor positioning and an attitude-determination method. The difference between the round trip time (RTT) ranging information corrected with a standard time deviation and the distance information calculated with an INS was used as measurement information. NLOS errors were eliminated according to the set threshold, and indoor position and attitude determination were conducted through an extended Kalman filter. Reference [5] proposes an adaptive robust Kalman filtering method that uses the threshold to construct a robust factor that identifies and weakens NLOS ranging errors. Meanwhile, the same study used a Sage Husa filter to estimate and correct system noise in real time in order to improve the accuracy of the UWB positioning. Another study in the Reference [6], analyzed the characteristics of the clock that is offset of the UWB antenna in the actual environment, the relative speed between the nodes and the ranging errors caused by the non line of sight environment. Meanwhile, the same study constructed an error-compensation method to achieve positioning and improved the ranging and positioning accuracy of UWB positioning in practical applications. A third study in the literature, [7], estimated location information based on Chan and Taylor's collaborative location method via setting a threshold value for the residual of estimated results to identify the NLOS environment, calculating location results using the Kalman method for qualified measurement data, weighting the residual and moving averages of the location results and completing an update of the final location. Reference [8] proposed a robust-volume Kalman filter (CKF) algorithm with a noise–time-varying estimator. The robust equivalent covariance matrix was constructed via use of the predicted residual factor to control the influence of observation outliers on the filter-parameter solution. The Sage Husa algorithm was used to estimate and correct the statistical characteristics of system noise in real time, improve filtering accuracy and stability and achieve high-precision positioning. Reference [9] proposed a robust Student's t-based Kalman filter, which provided a Gaussian approximation of posterior distribution.

In order to avoid deviation of UWB position estimation caused by linearization, some researchers used an unscented Kalman filter (UKF) for UWB position estimation, estimated the position through traceless transformation of sigma points and obtained the UWB position estimation algorithm based on the UKF [10,11]. However, the algorithms based on the extended Kalman filter (EKF) and the UKF both assume that the state noise and the measurement noise will obey Gaussian distribution; therefore, they are not suitable for non Gaussian noise. However, in the UWB indoor-positioning process, non Gaussian noise, such as the noise of mobile cars, pulse noise caused by electromagnetic interference and communication system faults or defects, does exist objectively [12,13].

Aiming at the problem of measurement-information abnormal-error and nonlinear filtering for UWB positioning, this paper proposes a maximum cross-correlation entropy-based unscented Kalman filter algorithm based on the maximum cross-correlation criterion on the basis of a nonlinear filter algorithm. The algorithm first obtains a predictive state estimate and a covariance matrix through traceless transformation and then reconstructs observation information using the nonlinear regression method based on the maximum cross-correlation entropy criterion, which enhances robustness of the unscented Kalman filter algorithm.

This paper is arranged as follows: Section 2 introduces the UWB positioning method based on nonlinear filtering, including the principle of UWB positioning and the UWB positioning method based on the unscented Kalman filter. Section 3 introduces the pro-

posed maximum cross-correlation entropy UKF algorithm, including the maximum cross-correlation entropy criterion and the proposed improved UKF algorithm. In Section 4, the effectiveness of the proposed method is demonstrated through simulation and an experiment. The fifth section is the conclusion.

## 2. UWB Positioning Method Based on Unscented Kalman Filter

### 2.1. Principle of UWB Positioning

At present, common range-based wireless positioning methods include the signal-arrival time-based method, the signal-arrival time-difference-based method, the signal-arrival angle-based method and the received-signal strength-based method [14–16]. UWB technology is carrier-free communication technology. It uses narrow nanosecond nonsinusoidal pulses to transmit data. It can measure precise signal transmission time, which can accurately measure the distance between the base station and the label. Therefore, UWB technology usually adopts a time of arrival (TOA)-based localization algorithm.

The TOA positioning method is also called the circle positioning method, which calculates the coordinates of a position label by finding the intersection points of circles [17]. The location of the base station (BS) is used as the center of each circle, and the distance from the BS to the location label is used as the radius to draw each circle. If there is no error in the distances, the final circles will intersect at a point, which is the label position.

Based on the TOA model, a schematic diagram of the positioning principles of three base stations and one label is established on a plane. The base-station coordinate is defined as $BS(x_i, y_i)$, and the coordinates of the positioning label are Label $(x, y)$. The measured distance from the base station to the label is $R_i$. The distance with the base station as the center, $R_i$ is the radius, which forms the circle-position line. Due to the existence of ranging errors, the positioning model is shown in Figure 1.

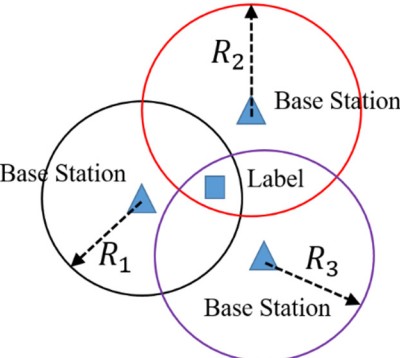

**Figure 1.** TOA-based geometric positioning model.

Since the coordinates, $BS(x_i, y_i)$, of the three base stations and the distance between the three base stations and the label are known, the relationship between the coordinate position and the distance can be established. Through the analysis above, the following equations can be obtained:

$$\begin{cases} (x_1 - x)^2 + (y_1 - y)^2 = R_1^2 \\ (x_2 - x)^2 + (y_2 - y)^2 = R_2^2 \\ (x_3 - x)^2 + (y_3 - y)^2 = R_3^2 \end{cases} \tag{1}$$

Calculate the difference between two pairs of the equations above to formulate

$$\begin{cases} R_2^2 - R_1^2 = (x_2^2 + y_2^2) - (x_1^2 + y_1^2) - 2(x_2 - x_1)x - 2(y_2 - y_1)y \\ R_3^2 - R_1^2 = (x_3^2 + y_3^2) - (x_1^2 + y_1^2) - 2(x_3 - x_1)x - 2(y_3 - y_1)y \end{cases} \tag{2}$$

The formula above is written in matrix form as follows:

$$\begin{bmatrix} x_2 - x_1 & y_2 - y_1 \\ x_3 - x_1 & y_3 - y_1 \end{bmatrix} \begin{bmatrix} x \\ y \end{bmatrix} = \frac{1}{2} \begin{bmatrix} (x_2^2 + y_2^2)^2 - (x_1^2 + y_1^2) + R_1^2 - R_2^2 \\ (x_3^2 + y_3^2)^2 - (x_1^2 + y_1^2) + R_1^2 - R_3^2 \end{bmatrix} \tag{3}$$

The coordinates of the label can be obtained from the formula above.

$$\begin{bmatrix} x \\ y \end{bmatrix} = \frac{1}{2} \begin{bmatrix} x_2 - x_1 & y_2 - y_1 \\ x_3 - x_1 & y_3 - y_1 \end{bmatrix}^{-1} \begin{bmatrix} (x_2^2 + y_2^2)^2 - (x_1^2 + y_1^2) + R_1^2 - R_2^2 \\ (x_3^2 + y_3^2)^2 - (x_1^2 + y_1^2) + R_1^2 - R_3^2 \end{bmatrix} \tag{4}$$

The position of the label can be obtained through solving Equation (4). However, in most cases, there are more than three base stations. Therefore, the least squares method (LSM) is usually used to solve Equation (4) so as to obtain a more accurate label position. The TOA method requires high clock synchronization between the labels and the BS. If the clock is not synchronized, the signal time will be incorrect when the label arrives at the BTS. If erroneous data is used for positioning, the positioning accuracy of the positioning system will decline, so in order to achieve high-precision positioning, the clock synchronization between the label and the BTS must be guaranteed, though that is difficult to achieve in practice.

### 2.2. UWB Location Algorithm Based on the UKF

Formula (4) obtains the initial positioning coordinates of unknown nodes. The UKF algorithm is modified according to the characteristics of the NLOS environment to perform precise positioning and position tracking. The UKF is a new nonlinear filter estimation algorithm that is UT (unscented)-based. The UKF and the EKF have different approaches to linearizing nonlinear functions. The UKF uses UT transformation to deal with nonlinear transference of the mean and the covariance, and directly uses the nonlinear model of the system. The EKF approximates nonlinear functions and needs the Jacobian matrix to be calculated, which greatly increases the amount of calculation. The UKF algorithm obtains more observational assumptions through generation of Sigma points and ignores higher-order terms without linearization. Therefore, in terms of nonlinear problems in an NLOS environment, the UKF has higher calculation accuracy and stronger adaptability than has the EKF [18].

The state equation is established as follows:

$$\dot{X} = FX + W \tag{5}$$

where $F$ is the state transition matrix and $W$ represents the matrix of noise. $X$ represents the state vector, which can be expressed as

$$X = \begin{bmatrix} P_x & P_y & V_x & V_y \end{bmatrix}^T \tag{6}$$

where $P_x$ and $P_y$ represent the positioning coordinates of the label in the X direction and the Y direction, respectively. $V_x$ and $V_y$ indicate the motion velocity of the label in the X and Y directions, respectively. $F$ from Equation (5) is

$$F = \begin{bmatrix} 1 & 0 & \Delta T & 0 \\ 0 & 1 & 0 & \Delta T \\ 0 & 0 & 1 & 0 \\ 0 & 0 & 0 & 1 \end{bmatrix} \tag{7}$$

According to the matrix of noise, the covariance matrix, Q, is

$$Q = q \begin{bmatrix} \Delta T^3/3 & 0 & \Delta T^2/2 & 0 \\ 0 & \Delta T^3/3 & 0 & \Delta T^2/2 \\ \Delta T^2/2 & 0 & \Delta T & 0 \\ 0 & \Delta T^2/2 & 0 & \Delta T \end{bmatrix} \tag{8}$$

where q is the power spectrum density of the system noise. $\Delta T$ is the UWB-data sampling interval. According to the relationship between the position and the distance of the two coordinate points, the equation for the distance between the label and the base station can be established. Considering the existence of multiple base stations, the following general equation is given:

$$R_{i,k} = \sqrt{(x_k - x_i)^2 + (y_k - y_i)^2} + n_{i,k} \tag{9}$$

where $R_{i,k}$ represents the measurement distance and $n_{i,k}$ is measurement noise. When the viewing-distance environment is between the base station and the mobile station, $n_{i,k}$ follows zero-mean Gaussian distribution. When the base station and mobile station are in a non line of sight environment or abnormal ranging occurs, then the measurement equation is

$$Z = HX + V \tag{10}$$

where H represents the measurement transfer matrix and V represents measurement noise. Measurement matrix Z consists of the distance measured with UWB positioning:

$$Z = [R_{1,k} \ R_{2,k} \ \cdots \ R_{M,k}]^T \tag{11}$$

$$H = \begin{bmatrix} \sqrt{(x_k - x_1)^2 + (y_k - y_1)^2} \\ \sqrt{(x_k - x_2)^2 + (y_k - y_2)^2} \\ \cdots \\ \sqrt{(x_k - x_M)^2 + (y_k - y_M)^2} \end{bmatrix} \tag{12}$$

$$V = [n_{1,k} \ n_{2,k} \ \cdots \ n_{M,k}]^T \tag{13}$$

To sum up, the steps of the UWB positioning algorithm based on the UKF are as follows:
Step 1: Set the initial parameters of the UKF filter:

$$\begin{cases} \hat{X}_0 = E(X_0) \\ P_0 = E\left[(X - X_0)(X - X_0)^T\right] \end{cases} \tag{14}$$

Step 2: According to Equation (14), calculate the sigma sample points:

$$\chi_{k-1} = \left[\hat{X}_{k-1}\hat{X}_{k-1} + \gamma\sqrt{P_{k-1}}\hat{X}_{k-1} - \gamma\sqrt{P_{k-1}}\right]^T \tag{15}$$

Among them, $P_{k-1}$ represents the covariance matrix, $\chi_{k-1}$ represents the composed column vector and $\gamma$ represents a scale factor.
Step 3: Calculate the covariance matrix and the predicted state vector:

$$\begin{cases} \chi_{k|k-1} = f(\chi_{k-1}) \\ \hat{X}_{k|k-1} = \sum_{i=0}^{2n} w_i^m \chi_{k|k-1}^i \\ P_{k|k-1} = \sum_{i=0}^{2n} w_i^c \left[\chi_{k|k-1}^i - \hat{X}_{k|k-1}\right] \end{cases} \tag{16}$$

Among these variables, $w_i^m$ is the weight of the mean value and $w_i^c$ is the weight of the covariance.

Step 4: With $\hat{X}_{k|k-1}$ and $P_{k-1}$ as the output values, calculate the sigma sample points according to Step 2 ($\zeta_{k-1}$), then calculate the measurement information predicted at time $k$:

$$\zeta_{k-1} = \left[ \hat{X}_{k|k-1} \hat{X}_{k|k-1} + \gamma \sqrt{P_{k|k-1}} \hat{X}_{k|k-1} - \gamma \sqrt{P_{k|k-1}} \right]^T \tag{17}$$

Step 5: Measure and update to obtain the covariance matrix and the predicted state vector:

$$\begin{cases} \zeta_{k|k-1} = h(\zeta_{k-1}) \\ \hat{Z}_{k|k-1} = \sum_{i=0}^{2n} w_i^m \zeta_{k|k-1}^i \end{cases} \tag{18}$$

$$P_{xz} = \sum_{i=0}^{2n} w_i^c \left[ \chi_{k|k-1}^i - \hat{X}_{k|k-1} \right] \left[ \chi_{k|k-1}^i - \hat{X}_{k|k-1} \right]^T \tag{19}$$

$$P_{zz} = \sum_{i=0}^{2n} w_i^c \left[ \zeta_{k|k-1}^i - \hat{Z}_{k|k-1} \right] \left[ \zeta_{k|k-1}^i - \hat{Z}_{k|k-1} \right]^T + R_k \tag{20}$$

$$\begin{cases} K_k = P_{xz} P_{zz}^{-1} \\ \hat{X}_k = \hat{X}_{k-1} + K_k \left[ Z_k - \hat{Z}_{k|k-1} \right] \\ P_k = P_{k|k-1} - K_k P_{zz} K_k^T \end{cases} \tag{21}$$

where $K_k$ is the gain matrix, $\hat{X}_k$ is the state vector at time $k$, and $P_k$ is the covariance matrix at time $k$.

## 3. Proposed Algorithm Based on the Maximum-Correntropy UKF

In this paper, based on the UKF, we first obtained the predicted state information and the covariance matrix through traceless transformation. Then, the nonlinear regression method based on the maximum cross-correlation entropy criterion was used to reconstruct observation information, which enhanced the robustness of the unscented Kalman filter algorithm for heavy-tailed noise.

### 3.1. Principle of Maximum Correntropy

The maximum entropy principle, also known as the maximum information principle, is a criterion for selecting the statistical characteristics of random variables that are the most consistent with an objective situation. The probability distribution of a random quantity is difficult to measure, and generally, only its various mean values, such as mathematical expectation and variance, or the values known under certain conditions. The distribution of these values can be varied or even infinite. Generally, one distribution will have the largest entropy. We select the distribution with the maximum entropy as the distribution of the random variable is an effective method and criterion [19].

Given two random variables, $X$ and $Y$, the correlation entropy between them is defined as [20,21]

$$V(X, Y) = E[\kappa(X, Y)] = \iint \kappa(X, Y) F(X, Y) dx dy \tag{22}$$

where $E[\cdot]$ represents the expectation, $\kappa(X, Y)$ represents the kernel function and $F(X, Y)$ represents the joint probability density function of the random variables $X$ and $Y$. The authors of this paper selected the Gaussian kernel function as their kernel function:

$$\kappa(x_i, y_i) = G_\sigma(e_i) = exp\left( \frac{e_i^2}{2\sigma^2} \right) \tag{23}$$

where $\{x_i, y_i\}_{i=1}^{N}$ represents N samples meeting the joint probability density, $F(X, Y)$. $e_i = x_i - y_i$. $\sigma$ represents the bandwidth of the kernel function.

Because only a limited part of the data in the actual system is known and it is difficult to accurately obtain joint probability density, the average value of the sample is usually used to calculate the estimated value of the correlation entropy:

$$V(\boldsymbol{X}, \boldsymbol{Y}) = \frac{1}{N} \sum_{i=1}^{N} \kappa(x_i, y_i) \tag{24}$$

Through substitution of Equation (23) into Equation (24) and expanding of the Gaussian kernel function with the Taylor expansion, we derived

$$V(\boldsymbol{X}, \boldsymbol{Y}) = \sum_{n=0}^{\infty} \frac{(-1)^n}{2^n \sigma^{2n} n!} E\left[(\boldsymbol{X} - \boldsymbol{Y})^{2n}\right] \tag{25}$$

The above formula shows that correlation entropy is obtained from the weighted sum of all even-order moments of the random variable $(\boldsymbol{X} - \boldsymbol{Y})$. Therefore, Formula (25) contains high-order moment information. At the same time, the correlation entropy, $V(\boldsymbol{X}, \boldsymbol{Y})$, receives the maximum value of 1 only when $\boldsymbol{X} = \boldsymbol{Y}$. With selection of the right σ, higher-order moments can be incorporated into the signal-processing algorithm to more accurately describe the higher-order characteristics of the non Gaussian distribution. This is the main advantage of correlation entropy in processing non Gaussian noise.

### 3.2. Proposed Algorithm

This section uses the UWB positioning system as the research object. Aiming at the problem of outlier interference in NLOS environments, a novel unscented Kalman filter algorithm was designed with the maximum cross-correlation entropy criterion and the introduction of the Gaussian kernel function. The specific derivation process is as follows:

Combined with the maximum-entropy criterion and the MMSE criterion, the recursive expression of the UKF can be obtained via solving the following cost function:

$$J_{KF} = \min_{\boldsymbol{X}_k} \left( \|\boldsymbol{X}_k - \hat{\boldsymbol{X}}_{k|(k-1)}\|^2_{\boldsymbol{P}^{-1}_{k|(k-1)}} + \|\boldsymbol{H}_k \boldsymbol{X}_k - \boldsymbol{Z}_k\|^2_{\boldsymbol{R}^{-1}_k} \right) \tag{26}$$

where $\hat{\boldsymbol{X}}_{k|(k-1)}$ represents the state vector, $\boldsymbol{X}_k$ is the a priori estimate of $k$ and $\boldsymbol{P}_{k|(k-1)}$ represents the prior error covariance matrix;

$$\boldsymbol{e}_k = \boldsymbol{R}_k^{-1/2}(\boldsymbol{H}_k \boldsymbol{X}_k - \boldsymbol{Z}_k) \tag{27}$$

The above formula can be expressed as

$$J_{KF} = \min_{\boldsymbol{X}_k} \left( \|\boldsymbol{X}_k - \hat{\boldsymbol{X}}_{k|(k-1)}\|^2_{\boldsymbol{P}^{-1}_{k|(k-1)}} + \sum_{i=1}^{n} e_{ki}^2 \right) \tag{28}$$

where $e_{ki}$ is the component of $k$ is $i = (1, \cdots, m)$. The formula above shows that under the MMSE criterion, the cost function of the Equation (28) assigns equal weight values to the residuals of all observations. However, the optimal estimation result cannot be obtained in the case of a measurement anomaly. Through introduction of the Gaussian kernel function in the maximum cross-correlation entropy criterion into the measurement of the cost function, the cost function of Equation (28) can be modified as follows:

$$J_{MCKF} = \min_{\boldsymbol{X}_k} \left( \|\boldsymbol{X}_k - \hat{\boldsymbol{X}}_{k|(k-1)}\|^2_{\boldsymbol{P}^{-1}_{k|(k-1)}} + \sigma^2 \sum_{i=1}^{n} G_\sigma(e_{ki}) \right) \tag{29}$$

Among these variables, $\sigma > 0$ indicates the bandwidth of the kernel function. Formula (29) replaces the MMSE estimation in the measurement-error section of the UKF with the kernel function of the maximum cross-correlation entropy.

For Formula (29), in $X_k$, the derivation at $k$ gives:

$$P_{k|(k-1)}^{-1}\left(X_k - \hat{X}_{k|(k-1)}\right) + \sum_{i=1}^{n} G_\sigma(e_{ki})e_{ki}\frac{\partial e_{ki}}{\partial x_k} = 0 \tag{30}$$

The formula above is organized into matrix form as follows:

$$P_{k|(k-1)}^{-1}\left(X_k - \hat{X}_{k|(k-1)}\right) + H_k^T R_k^{-T/2}\psi_k e_k = 0 \tag{31}$$

Among these variables, $\psi_k = diag[G_\sigma(e_{ki})]$ indicates the weight matrix and $diag[\bullet]$ represents a diagonal matrix. Through changing of $e_k$ via substitution of the expression of k into Formula (31), we derived

$$P_{k|(k-1)}^{-1}\left(X_k - \hat{X}_{k|(k-1)}\right) + H_k^T R_k^{-T/2}\psi_k R_k^{-1/2}(H_k X_k - Z_k) = 0 \tag{32}$$

The above equation can be regarded as Equation (33) for the $X_k$ derivative at $k$:

$$J_{MCKF} = \min_{X_k}\left(\|X_k - \hat{X}_{k|(k-1)}\|_{P_{k|(k-1)}^{-1}}^2 + \|H_k X_k - Z_k\|_{\tilde{R}_k^{-1}}^2\right) \tag{33}$$

where

$$\tilde{R}_k^{-1} = R_k^{-T/2}\psi_k R_k^{-1/2} \tag{34}$$

Comparison of Formula (28) and Formula (33) shows that the filtering algorithm based on the maximum cross-correlation entropy passes the weight matrix on the basis of the unscented Kalman filtering algorithm, $\psi_k$. The covariance of measurement noise is modified. The MCUKF method is as follows:

Step 1: Initialize the state vector and the covariance matrix.
Step 2: State one-step forecast update.
Step 3: State the one-step prediction mean square error update.
Step 4: According to Formula (34), update the measurement-noise covariance matrix.
Step 5: According to Formula (21), update the UKF measurement.

## 4. Results and Analysis

To verify the UKF algorithm based on the maximum cross-correlation entropy criterion proposed in this paper, a simulation and experimental verification were designed.

### 4.1. Simulation

The UWB positioning simulation platform include four base stations and one label, which was designed according to the UWB positioning principle. The local position frame, in which the positioning coordinates of the four base stations were BS1 (1 m,1 m), BS2 (14 m, 2 m), BS3 (16 m, 19 m) and BS4 (2 m, 21 m), respectively, was established. In terms of setting the ranging-error parameters, the distance measureing error from the base station to the label was ±15 cm. In the simulation process, labels moved uniformly in a straight line within the area enclosed in the four base stations. The starting position of the label was (1 m, 10 m), and the whole process lasted for 500 s. Details are shown in Figure 2. In order to demonstrate the effectiveness of the MCUKF method, the least squares method (LSM) was introduced to compare with the CKF.

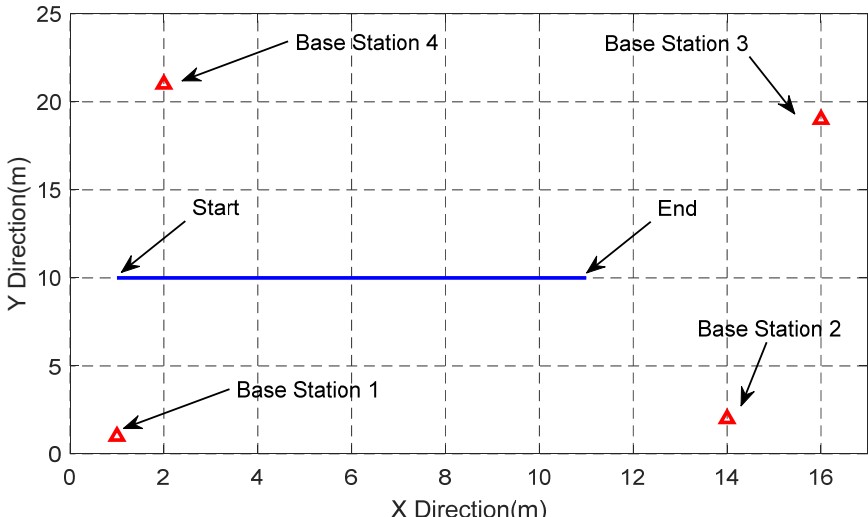

**Figure 2.** Simulation of carrier-movement track and base-station distribution.

Based on the simulation conditions above, the error curve of the positioning results was drawn, as shown in Figure 3. The LSM is marked with a black dashed line, the UKF method is marked with a blue line and the MCUKF method is marked with a red line. As can be seen from the figure, the MCUKF method proposed in this paper is superior to the LSM and the UKF method in positioning errors in both the X direction and the Y direction.

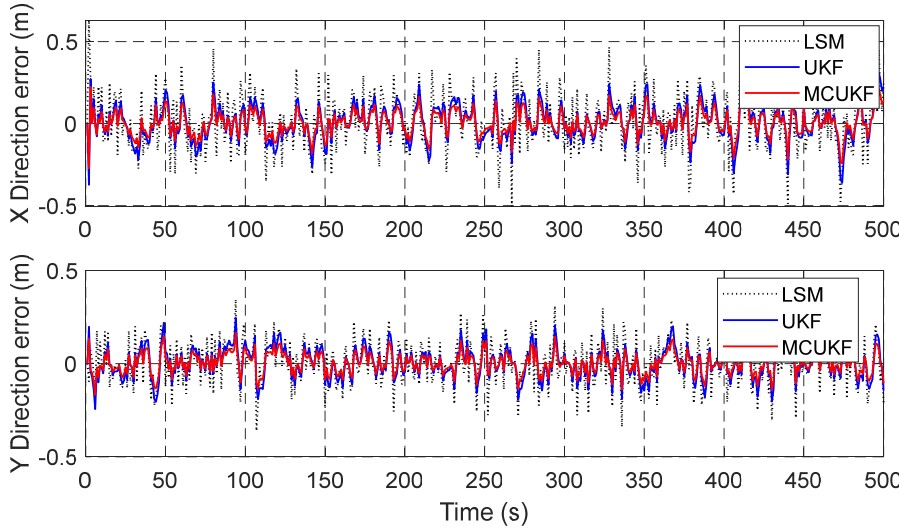

**Figure 3.** X− and Y−direction position-error simulation curve based on MCUKF method.

In order to comprehensively compare the positioning accuracy of the three methods, the horizontal position error curve, shown in Figure 4, was drawn. This figure shows that the horizontal positioning accuracy of the MCUKF method was better than that of the LSM and the UKF method.

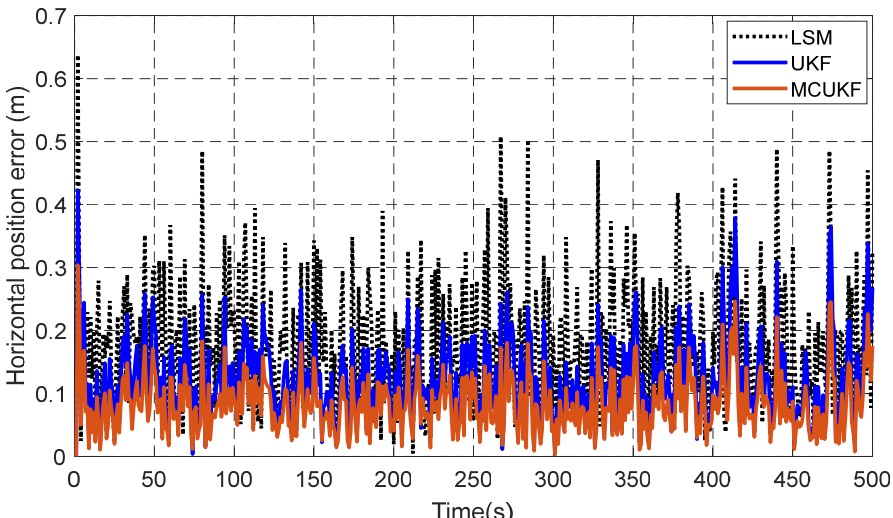

**Figure 4.** Horizontal position-error simulation curve based on MCUKF method.

The results in Figures 3 and 4 gave the statistical table of the root mean square error shown in Table 1. This table shows that the MCUKF method is superior to the LSM and the UKF method no matter the root mean square error of the X or Y direction or the root mean square error of the horizontal position. Compared with the LSM, the root mean square error results of the X direction, the Y direction and the horizontal position found with the MCUKF method decreased by 54.50%, 52.52% and 53.78%, respectively. Compared with the UKF method, the MCUKF method values were reduced by 28.89%, 31.76% and 31.34%, respectively. Therefore, the proposed MCUKF method has better positioning accuracy.

**Table 1.** Simulation of RMS of UWB positioning errors.

| Method | X-Direction Error (m) | Y-Direction Error (m) | Horizontal Position Error (m) |
|--------|-----------------------|-----------------------|-------------------------------|
| LSM | 0.1655 | 0.1190 | 0.2038 |
| UKF | 0.1059 | 0.0828 | 0.1372 |
| MCUKF | 0.0753 | 0.0565 | 0.0942 |

*4.2. Test Verification*

To verify the MCUKF method, a UWB positioning system based on the base station and the label was designed. The system included four base stations, one label, a computer and a trolley; the specific block diagram is shown in Figure 5. The computer was responsible for recording information about the distance from the label to the four base stations, as well as the positioning results.

The test system was built as shown in Figure 6, according to the framework above. The test area was an underground garage. One of the base stations was taken as the coordinate origin, and the four UWB base stations were each placed in a different corner. The label was placed on the trolley, and the trolley ran according to the specified route. The UWB update frequency was 2 Hz.

The whole test was divided into two parts: a static test and a dynamic test. It mainly verified the performance of the MCUKF algorithm under external interference. Figure 7 shows the distribution of the positioning results of the three methods when the label was stationary. The figure shows that due to the influence of external interference, the positioning results of the least squares method and the UKF method fluctuated. The MCUKF method reduced the external interference and had high positioning accuracy.

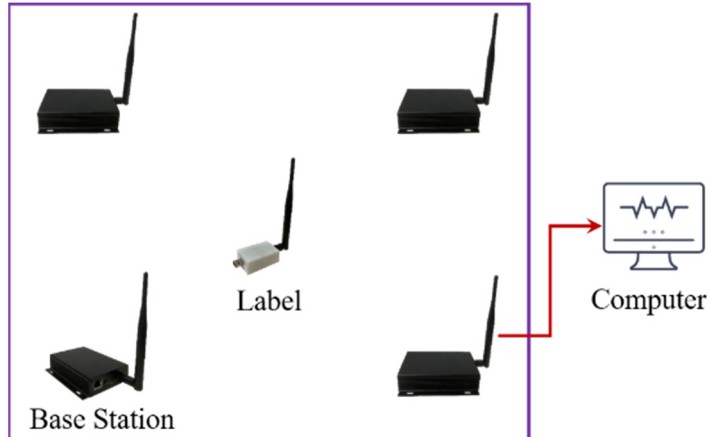

**Figure 5.** Block diagram of UWB positioning system.

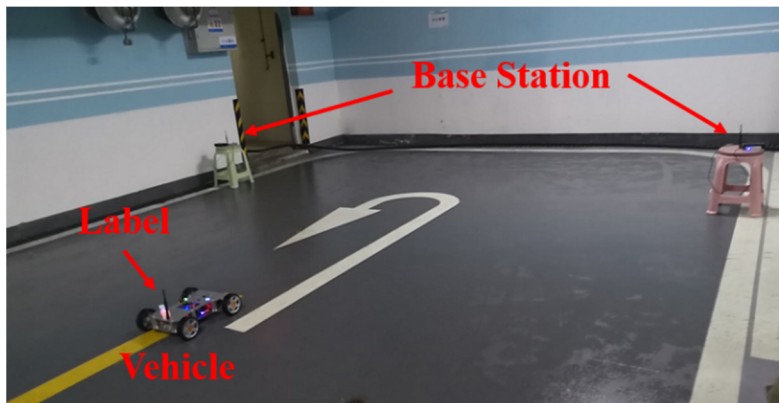

**Figure 6.** Field-test diagram of UWB positioning system.

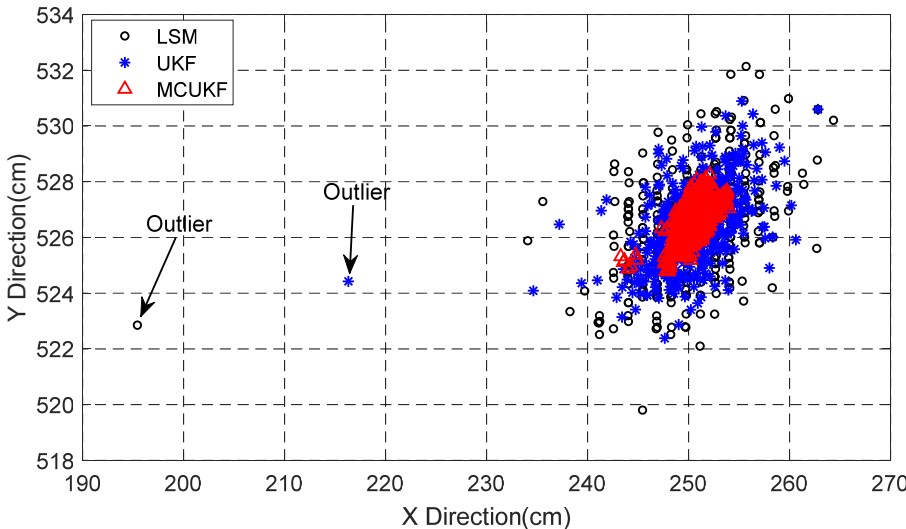

**Figure 7.** Comparison of positioning results of different methods under static conditions.

To facilitate analysis of the above results, the distribution curve of the standard deviation of the positioning results was drawn, as shown in Figure 8. The standard-deviation curves of the positioning results in the X and Y directions in the figure show that the standard deviation of the positioning results in the lowest multiplication method was the largest. The MCUKF method presented in this paper had the minimum standard deviation distribution.

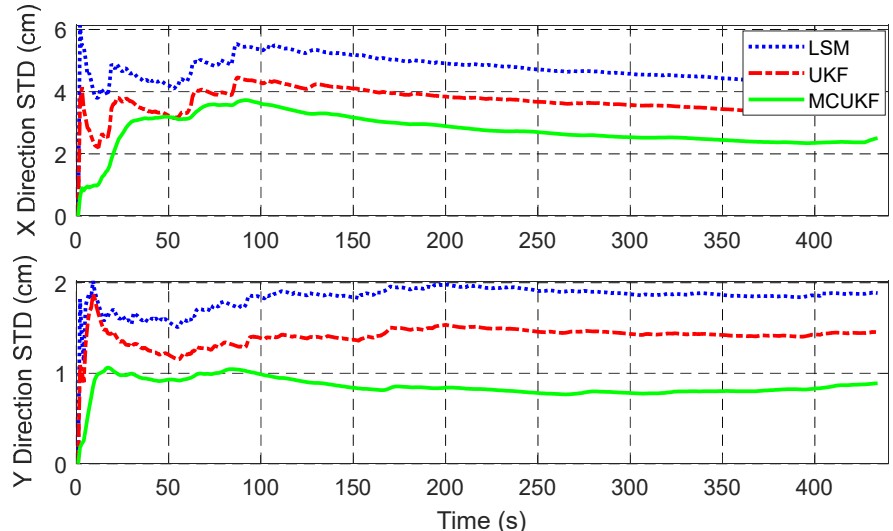

**Figure 8.** Distribution curve of standard deviation of positioning results.

Statistical analysis of the results in Figure 8 obtained the average-standard-deviation statistical results shown in Table 2. The average standard deviations of the MCUKF method proposed in this paper, in the X direction and the Y direction, were 1.6075 m and 0.7472 m, respectively. As seen in this table, the MCUKF method had the smallest mean standard deviation compared with the LSM and the UKF method. This table also shows that the proposed UWB localization method based on the MCUKF method is robust.

**Table 2.** Statistical results of mean standard deviation.

| Method | X-Direction Error (m) | Y-Direction Error (m) |
|--------|----------------------|----------------------|
| LSM | 5.1034 | 1.8855 |
| UKF | 3.9037 | 1.4531 |
| MCUKF | 1.6075 | 0.7472 |

In order to illustrate the universal applicability of this method, the static-positioning test was redesigned. Figure 9 shows the UWB positioning results under the condition of double interference. The figure shows that when there was external interference, the positioning results of the least squares method and the UKF method both had outlier fluctuations. The MCUKF method proposed in this paper still maintained high robustness and positioning accuracy.

Figure 10 is the standard-deviation distribution curve under the condition of interference drawn on the basis of Figure 9. This figure shows that the standard-deviation curve also fluctuated greatly due to the influence of large outliers. The standard-deviation curve of the MCUKF method proposed in this paper remained stable under the influence of large outliers.

Similarly, the average standard-deviation statistical results of the different methods shown in Table 3 are listed according to the results in the figure above. The average standard deviations of the MCUKF method were 1.661 m in the X direction and 0.752 m in the Y direction, respectively. Meanwhile, compared with those of the LSM, the mean standard deviations of the method were reduced by 74.85% and 67.03%, respectively. Compared withUKF method, the average standard deviations decreased by 63.14% and 54.51%, respectively. The MCUKF method had high robustness.

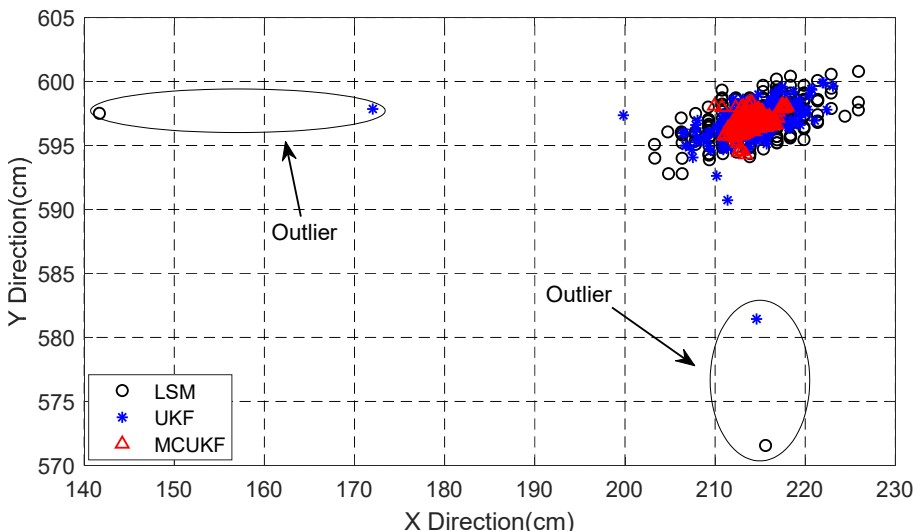

**Figure 9.** Test chart of UWB positioning results under static conditions.

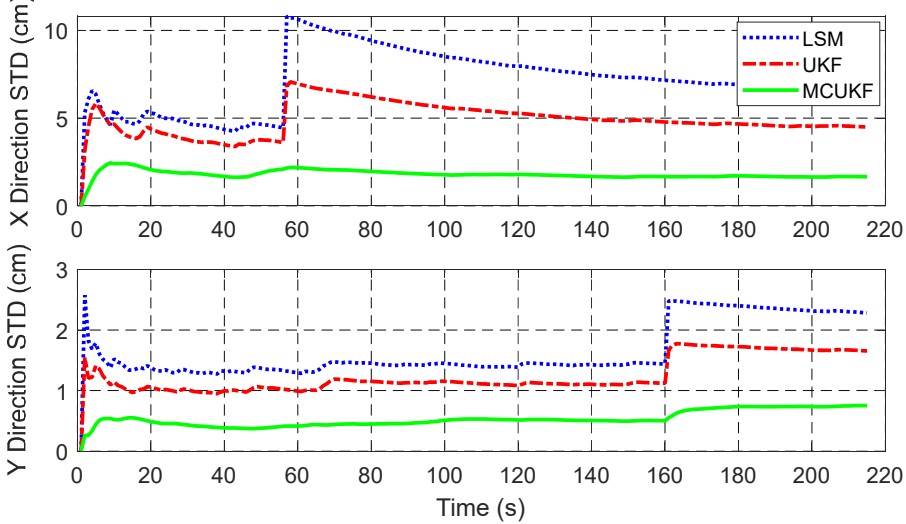

**Figure 10.** Distribution curve of standard deviation under interference.

**Table 3.** Statistical results of mean standard deviation for different methods.

| Method | X-Direction Error (m) | Y-Direction Error (m) |
|:---:|:---:|:---:|
| LSM | 6.603 | 2.281 |
| UKF | 4.506 | 1.653 |
| MCUKF | 1.661 | 0.752 |

In order to further illustrate the effectiveness of the proposed method, a dynamic UWB positioning test was designed. The moving area of the trolley was 14 m × 5 m, including acceleration, deceleration, straight lines and turning.

Figure 11 shows the positioning-result error curves of the three methods under dynamic conditions. The circle symbol represents the least squares method. The blue line represents the UKF method. The red line represents the MCUKF method proposed in this paper. This figure shows that the positioning result of the least squares method had large fluctuations due to external interference. Meanwhile, compared with the LSM, the positioning result of the UKF method was relative, but still displayed fluctuation. The MCUKF method proposed in this paper had the best robustness and high positioning accuracy.

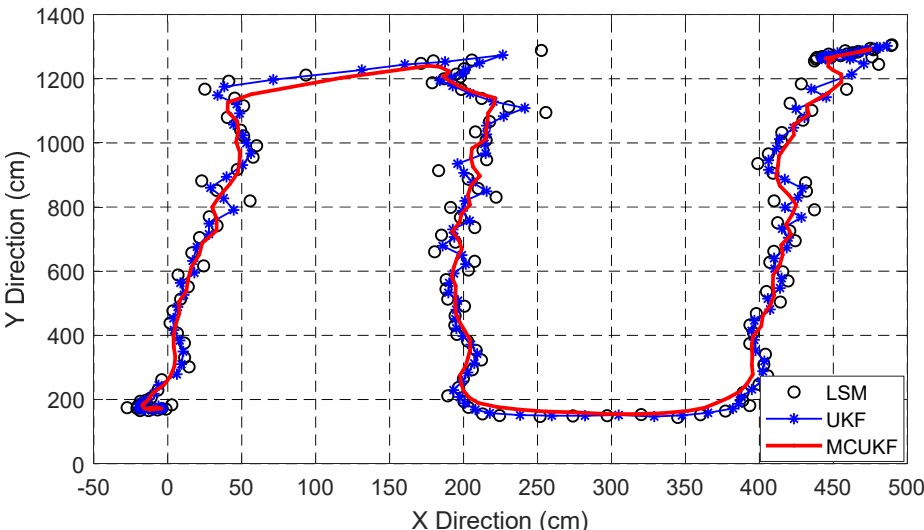

**Figure 11.** Curves of positioning results of various methods under dynamic conditions.

**5. Conclusions**

Aiming at the problem that UWB location is easily interfered with by outliers, this paper proposes a UKF algorithm based on maximum cross-correlation entropy. On the one hand, a UWB location model based on nonlinear filtering was derived. On the other hand, the predicted state estimate and the covariance matrix were obtained through traceless transformation, and the observation information was reconstructed using the nonlinear regression method based on the maximum cross-correlation entropy criterion to further improve the anti-interference ability of the filter. The simulation and vehicle test results show that the MCUKF algorithm proposed in this paper can maintain high robustness under the condition of wild value interference. At the same time, the UWB positioning accuracy of the algorithm proposed in this paper is better than that of the least squares method and the UKF method.

**Author Contributions:** Writing—original draft, M.Z.; Writing—review & editing, T.Z. and D.W. All authors have read and agreed to the published version of the manuscript.

**Funding:** This work was supported in part by the Key R&D program of Jiangsu Province under Grant BE2021679, the Key R&D and Transformation program of Qinghai Province under Grant 2022-QY-208 and the National Disabled Persons' Federation project under Grant 2021CDPFAT-26.

**Institutional Review Board Statement:** Not applicable.

**Informed Consent Statement:** Not applicable.

**Data Availability Statement:** Not applicable.

**Conflicts of Interest:** The authors declare no conflict of interest.

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
