# Peer review of "A Novel UWB Positioning Method Based on a Maximum-Correntropy Unscented Kalman Filter"

_applsci, doi:10.3390/app122412735_

Round 1
Reviewer 1 Report
this paper proposed a modified KF to improve the UWB position accuracy, which is interesting.
the introducton is insufficient, please cite following papers:
Jiang, C., Shen, J., Chen, S., Chen, Y., Liu, D., & Bo, Y. (2020). UWB NLOS/LOS classification using deep learning method. IEEE Communications Letters, 24(10), 2226-2230. Jiang, C., Chen, S., Chen, Y., Liu, D., & Bo, Y. (2020). An UWB channel impulse response de-noising method for NLOS/LOS classification boosting. IEEE Communications Letters, 24(11), 2513-2517. Factor graph optimization is a new method which has been demonstrated superior than KF, it is interesting to explore FGO-UWB.
Author Response
Original Manuscript ID: applsci-2057368
Original Article Title: “A Novel UWB Position Method Based on Maximum Correntropy Unscented Kalman Filter”
Reviewer#1, Concern # 1: The introducton is insufficient, please cite following papers: Jiang, C., Shen, J., Chen, S., Chen, Y., Liu, D., & Bo, Y. (2020). UWB NLOS/LOS classification using deep learning method. IEEE Communications Letters, 24(10), 2226-2230. Jiang, C., Chen, S., Chen, Y., Liu, D., & Bo, Y. (2020). An UWB channel impulse response de-noising method for NLOS/LOS classification boosting. IEEE Communications Letters, 24(11), 2513-2517. Factor graph optimization is a new method which has been demonstrated superior than KF, it is interesting to explore FGO-UWB.
Author response: Thank you very much for your suggestion. We agree with your point of view.
Author action: According to the requirements of the reviewer, we have added relevant references in the introduction section.
- Jiang C, Shen J, Chen S, et al. UWB NLOS/LOS classification using deep learning method[J]. IEEE Communications Letters, 2020, 24(10): 2226-2230.
- Jiang C, Chen S, Chen Y, et al. An UWB channel impulse response de-noising method for NLOS/LOS classification boosting[J]. IEEE Communications Letters, 2020, 24(11): 2513-2517.
Reviewer 2 Report
This paper proposes ultra-wideband positioning algorithm based on maximum cross correlation entropy unscented Kalman filtering. Although the paper provided interesting analysis, experimental demonstration and results, an extensive editing of English language and style is required. Quality of presentation must be improved before the paper is recommended for publication. Points that need a particular consideration are as follows:
Abbreviations must be defined at their first appearance in the manuscript, for example, GNSS, INS, NLOS, TOA, BTS, etc.
In section 2.1 paragraph 2, the paper interchangeably referred to the positioning target by tag/label. In section 4.1 paragraph 1, the paper used “tag”, while in section 4.2 it used Label. To avoid any confusion that this may cause to the reader, it is recommended to use either tag or label.
All equations’ parameters must be defined at their first appearance. For example, in equation (6) what do P and V refer to? What is the difference between P in (6) and P in (15) and (16)? Similarly, Q in (8) was defined twice (as covariance matrix in line 152 and as power spectral density in line 153), while q was not defined.
In section 4.1 paragraph 2, both figure 2 and figure 3 was confusingly discussed. It is highly recommended to explain each figure separately first then compare between them if required.
In section 4.1, What is the difference between least square method in line 258 and lowest multiplication method in line 310?
Please proofread for grammar errors particularly, discussion redundancy, punctuations, and uncompleted sentences.
Author Response
Original Manuscript ID: applsci-2057368
Original Article Title: “A Novel UWB Position Method Based on Maximum Correntropy Unscented Kalman Filter”
Reviewer#2, Concern # 1: Abbreviations must be defined at their first appearance in the manuscript, for example, GNSS, INS, NLOS, TOA, BTS, etc.
Author response: Thank you very much for your suggestion. We agree with your point of view.
Author action: We have modified the relevant content as follows:
Global Navigation Satellite Systems (GNSS), Non-Line-Of-Sight (NLOS), Inertial Navigation System (INS), Round Trip Time (RTT), Unscented Kalman Filter (UKF), Extended Kalman Filter (EKF), Time-Of-Arrival (TOA), Base Station (BS)
Reviewer#2, Concern # 2: In section 2.1 paragraph 2, the paper interchangeably referred to the positioning target by tag/label. In section 4.1 paragraph 1, the paper used “tag”, while in section 4.2 it used Label. To avoid any confusion that this may cause to the reader, it is recommended to use either tag or label.
Author response: Thank you very much for your suggestion. We agree with your point of view.
Author action: We have modified the full text to uniformly use word label.
Reviewer#2, Concern # 3: All equations’ parameters must be defined at their first appearance. For example, in equation (6) what do P and V refer to? What is the difference between P in (6) and P in (15) and (16)? Similarly, Q in (8) was defined twice (as covariance matrix in line 152 and as power spectral density in line 153), while q was not defined.
Author response: Thank you very much for your suggestion. We agree with your point of view.
Author action: We have modified the relevant content as follows:
|
|
(6) |
Where and represent the position coordinates of the label in the x direction and the y direction respectively. and indicate the motion velocity of the label in the x and y directions, respectively.
According to the matrix of noise, the covariance matrix Q is:
|
|
(8) |
Where q is the power spectrum density of the system noise. ∆ T is the UWB data sampling interval.
Reviewer#2, Concern # 4: In section 4.1 paragraph 2, both figure 2 and figure 3 was confusingly discussed. It is highly recommended to explain each figure separately first then compare between them if required.
Author response: Thank you very much for your suggestion. We agree with your point of view.
Author action: We have modified the relevant content as follows:
According to the position principle of UWB, the UWB simulation position platform is designed, which including four base stations and one label. Establish the local position frame, in which the position coordinates of the four base stations are BS1(1m,1m), BS2(14m,2m), BS3(16m,19m) and BS4(2m,21m) respectively. In terms of setting the ranging error parameters, the ranging error from the base station to the label is ±15cm. In the simulation process, labels move uniformly in a straight line within the area enclosed by four base stations. The starting position of the label is (1m,10m), and the whole process lasts for 500s. The details are shown in Figure 2. In order to demonstrate the effectiveness of the MCUKF method, the least square method (LSM) was introduced to compare with CKF.
Figure 2. Simulation of carrier movement track and base station distribution.
Based on the above simulation conditions, the error curve of position results is drawn, as shown in Figure 3. The LSM method is marked by black dashed line, the UKF method is marked by blue line and the MCUKF method is marked by red line. As can be seen from the figure, the MCUKF method proposed in this paper is superior to LSM and UKF in both position errors in the X direction and Y direction.
In order to comprehensively compare the position accuracy of the three methods, the horizontal position error curve as shown in Figure 4 is drawn. It can be seen from the figure that the horizontal position accuracy of MCUKF method is better than LSM and UKF method.
Reviewer#2, Concern # 5: In section 4.1, What is the difference between least square method in line 258 and lowest multiplication method in line 310?
Author response: Thank you very much for your suggestion. We agree with your point of view.
Author action: Because of our handwriting mistakes. In fact, it should be the least square method.
Reviewer#2, Concern # 6: Please proofread for grammar errors particularly, discussion redundancy, punctuations, and uncompleted sentences.
Author response: Thank you very much for your suggestion. We agree with your point of view.
Author action: We have revised the grammar and other issues in the whole paper and marked them with yellow.

Reviewer 3 Report
Aiming at the problem that UWB location is easily interfered by outliers, this paper proposes a UKF algorithm based on maximum cross correlation entropy. The simulation and vehicle test results show that the MCUKF algorithm proposed in this paper can maintain high robustness under the condition of wild value interference. The paper has clear logic and detailed content. It is recommended to be accepted after minor revision. The reviewer’s comments are as follows.
1. In “1. Instruction”, many related works for addressing outliers are missing, such as “A novel robust Student's t-based Kalman filter”. The authors should add some discussions and comparisons illustrate the advantages of the present approach.
2. In section 3.2, the steps of MCUKF algorithm proposed in this paper are suggested.
3. In “Figure 5. Block Diagram of UWB Position System”, why is only one of the towers talking to the computer? It is necessary to give a specific test scheme framework.
4. There are a few grammatical mistakes in the paper, please correct them.
5. In “Figure 11. Curves of positioning results of various methods under dynamic conditions.”, why didn't the author give the reference position coordinates?
Author Response
Original Manuscript ID: applsci-2057368
Original Article Title: “A Novel UWB Position Method Based on Maximum Correntropy Unscented Kalman Filter”
Reviewer#3, Concern # 1: In “1. Instruction”, many related works for addressing outliers are missing, such as “A novel robust Student's t-based Kalman filter”. The authors should add some discussions and comparisons illustrate the advantages of the present approach.
Author response: Thank you very much for your suggestion. We agree with your point of view.
Author action: We have modified the relevant content as follows:
Reference [9] proposed a robust Student's t-based Kalman filter, which provides a Gaussian approximation to the posterior distribution.
Huang Y, Zhang Y, Li N, et al. A novel robust Student's t-based Kalman filter[J]. IEEE Transactions on Aerospace and Electronic Systems, 2017, 53(3): 1545-1554.
Reviewer#3, Concern # 2: In section 3.2, the steps of MCUKF algorithm proposed in this paper are suggested.
Author response: Thank you very much for your suggestion. We agree with your point of view.
Author action: We have modified the relevant content as follows:
The MCUKF is:
Step 1: Initialize the state vector and covariance matrix.
Step 2: UKF status one step forecast update.
Step 3: State one-step prediction mean square error update.
Step 4: According to formula (34), the measurement noise covariance matrix is updated.
Step 5: According to formula (21), UKF measurement was updated.
Reviewer#3, Concern # 3: In “Figure 5. Block Diagram of UWB Position System”, why is only one of the towers talking to the computer? It is necessary to give a specific test scheme framework.
Author response: Thank you very much for your suggestion. We agree with your point of view.
Author action: The UWB positioning system platform built in this paper includes 4 base stations, 1 label and 1 computer. Among them, the four base stations are divided into three sub-base stations and a host station. The host station is responsible for communicating with the computer, so there is only one base station connected to the computer.
Reviewer#3, Concern # 4: There are a few grammatical mistakes in the paper, please correct them.
Author response: Thank you very much for your suggestion. We agree with your point of view.
Author action: As required by the reviewer, we have revised the grammar of the whole paper.
Reviewer#3, Concern # 5: In “Figure 11. Curves of positioning results of various methods under dynamic conditions.”, why didn't the author give the reference position coordinates?
Author response: Thank you very much for your suggestion. We agree with your point of view.
Author action: To give a reference position you must provide a device with better positioning accuracy than UWB. Due to the limitation of experimental conditions, the equipment with better accuracy cannot be provided. Therefore, no reference position coordinates are given in Figure 11.
